

# The hypoferremic response to acute inflammation is maintained in thalassemia mice even under parenteral iron loading

Chanita Sanyear[1,2], Buraporn Chiawtada[3], Punnee Butthep[1], Saovaros Svasti[2], Suthat Fucharoen[2] and Patarabutr Masaratana[3]

[1] Department of Pathology, Faculty of Medicine Ramathibodi Hospital, Mahidol University, Bangkok, Thailand
[2] Thalassemia Research Center, Institute of Molecular Biosciences, Mahidol University, Nakhon Pathom, Thailand
[3] Department of Biochemistry, Faculty of Medicine Siriraj Hospital, Mahidol University, Bangkok, Thailand

Corresponding author
Patarabutr Masaratana,
patarabutr.mas@mahidol.ac.th

## ABSTRACT

**Background:** Hepcidin controls iron homeostasis by inducing the degradation of the iron efflux protein, ferroportin (FPN1), and subsequently reducing serum iron levels. Hepcidin expression is influenced by multiple factors, including iron stores, ineffective erythropoiesis, and inflammation. However, the interactions between these factors under thalassemic condition remain unclear. This study aimed to determine the hypoferremic and transcriptional responses of iron homeostasis to acute inflammatory induction by lipopolysaccharide (LPS) in thalassemic ($Hbb^{th3/+}$) mice with/without parenteral iron loading with iron dextran.

**Methods:** Wild type and $Hbb^{th3/+}$ mice were intramuscularly injected with 5 mg of iron dextran once daily for two consecutive days. After a 2-week equilibration, acute inflammation was induced by an intraperitoneal injection of a single dose of 1 μg/g body weight of LPS. Control groups for both iron loading and acute inflammation received equal volume(s) of saline solution. Blood and tissue samples were collected at 6 hours after LPS (or saline) injection. Iron parameters and mRNA expression of hepcidin as well as genes involved in iron transport and metabolism in wild type and $Hbb^{th3/+}$ mice were analyzed and compared by Kruskal–Wallis test with pairwise Mann–Whitney U test.

**Results:** We found the inductive effects of LPS on liver IL-6 mRNA expression to be more pronounced under parenteral iron loading. Upon LPS administration, splenic erythroferrone (ERFE) mRNA levels were reduced only in iron-treated mice, whereas, liver bone morphogenetic protein 6 (BMP6) mRNA levels were decreased under both control and parenteral iron loading conditions. Despite the altered expression of the aforementioned hepcidin regulators, the stimulatory effect of LPS on hepcidin mRNA expression was blunt in iron-treated $Hbb^{th3/+}$ mice. Contrary to the blunted hepcidin response, LPS treatment suppressed FPN1 mRNA expression in the liver, spleen, and duodenum, as well as reduced serum iron levels of $Hbb^{th3/+}$ mice with parenteral iron loading.

**Conclusion:** Our study suggests that a hypoferremic response to LPS-induced acute inflammation is maintained in thalassemic mice with parenteral iron loading in a hepcidin-independent manner.

## INTRODUCTION

Mammalian iron homeostasis mainly involves the adjustment of iron entry from intestinal iron absorption and reticuloendothelial (RE) iron recycling into the circulation to meet body iron demand, which is mainly dictated by erythropoietic activity. Dietary non-heme iron absorption involves many proteins, including divalent metal transporter 1 (DMT1), hephaestin, ferroportin (FPN1), and possibly duodenal cytochrome b (DCYTB) (*McKie et al., 2000*, *2001*; *Gunshin et al., 1997*; *Abboud & Haile, 2000*; *Donovan et al., 2000*; *Vulpe et al., 1999*). In RE iron recycling, iron is liberated from heme molecule in senescent erythrocytes by heme oxygenase enzyme. The resulting iron is transported into the cytoplasm by DMT1 (*Jabado et al., 2002*) and exported from the cells into the circulation by the iron efflux protein, FPN1 (*Donovan et al., 2005*). Hepcidin, the liver-secreted iron regulatory hormone, regulates systemic iron homeostasis by inducing the internalization and degradation of FPN1, thus, reducing intestinal iron absorption and RE iron recycling (*Nemeth et al., 2004b*; *Masaratana et al., 2011*). Hepcidin expression is regulated by multiple factors, including body iron stores, erythropoietic activity, hypoxia, and inflammation (*Nicolas et al., 2002a*, *2002b*; *Pigeon et al., 2001*). Hepcidin is upregulated in response to high body iron stores, whereas increased erythropoietic activity and ineffective erythropoiesis cause hepcidin suppression. Body iron stores regulate hepcidin via bone morphogenetic proteins (BMP)—particularly BMP6, which binds to BMP receptor and hemojuvelin (HJV), a BMP co-receptor, on the membrane of hepatocytes, and subsequently activates hepcidin transcription through Son of a mother against decapentaplegic (SMAD) signaling pathway (*Andriopoulos et al., 2009*; *Babitt et al., 2006*; *Wang et al., 2005*). Moreover, transferrin-bound iron can induce hepcidin expression via HFE and transferrin receptor 2 (TFR2) interaction (*Goswami & Andrews, 2006*; *Gao et al., 2009*).

Under hypoxia and iron deficiency, the expression of transmembrane serine protease 6 (TMPRSS6 or matriptase-2), which cleaves membrane-bound HJV, is increased leading to hepcidin suppression (*Du et al., 2008*; *Lakhal et al., 2011*). Additionally, iron and BMP6 can also activate TMPRSS6 expression, which possibly acts as a feedback mechanism to maintain appropriate hepcidin expression (*Meynard et al., 2011*).

Thalassemia, a hereditary blood disorder, is a significant global health problem. Beta-thalassemia is caused by mutations of gene encoding adult β-globin chains resulting in reduced β-globin synthesis and an imbalance between α and β-globin chains in erythroid cells. The precipitation of unmatched α-globin chains within erythroid cells results in erythroid cell destruction, ineffective erythropoiesis, extramedullary

hematopoiesis and anemia (*Higgs, Engel & Stamatoyannopoulos, 2012*). In the presence of ineffective erythropoiesis, erythroid regulators of hepcidin, namely growth differentiation factor 15 (GDF15), twisted gastrulation 1 (TWSG1), and particularly erythroferrone (ERFE), are released from erythroid precursor cells leading to hepcidin suppression (*Tanno et al., 2007*, *2009*; *Kautz et al., 2014*). Under thalassemic condition, intestinal iron absorption is enhanced; however, blood transfusion is needed in some patients to maintain hemoglobin levels, as well as to alleviate ineffective erythropoiesis and extramedullary hematopoiesis. The increased iron absorption along with blood transfusion consequently leads to systemic iron overload, which is one of the life-threatening complications of thalassemia. In addition to systemic iron overload, bacterial infection has been reported as one of the serious complications observed in thalassemic patients, especially after splenectomy (*Vento, Cainelli & Cesario, 2006*; *Ricerca, Di Girolamo & Rund, 2009*; *Teawtrakul et al., 2015*).

Inflammation and infection have been shown to transcriptionally induce hepcidin expression mainly through interleukin 6 (IL-6), which activates the Janus kinase/signal transducer and activator of transcription 3 (JAK/STAT3) signaling (*Nemeth et al., 2003*; *Maliken, Nelson & Kowdley, 2011*; *Rodriguez et al., 2014*). Inflammation-mediated hepcidin induction results in the suppression of FPN1 expression leading to reduced iron absorption and recycling, macrophage iron retention, and hypoferremia (*Nemeth et al., 2004a*). Additionally, hepcidin has been shown to play a role in defense mechanisms against siderophilic bacteria by reducing serum iron and NTBI levels in the host (*Stefanova et al., 2017*; *Arezes et al., 2015*).

It is noteworthy that interplay between different stimuli of iron homeostasis has been observed. It has been proposed that inductive effect of inflammation on hepcidin expression could be nullified by low iron status and erythropoietin drive (*Stoffel et al., 2019*). Therefore, the responses of hepcidin and iron parameters to acute inflammatory induction could be altered by iron status. A previous study reported that adequate hepatic iron content was required for hepcidin-mediated hypoferremic response to LPS challenge (*Fillebeen et al., 2018*). Furthermore, it was reported that low hepcidin was responsible for high inflammatory response to LPS in mice fed an iron deficient diet (*Pagani et al., 2011*). Moreover, the effect of inflammation on hepcidin expression could be affected by iron loading. In bone marrow-derived macrophages, LPS-induced hepcidin expression was suppressed by intracellular iron loading in a dose-dependent manner (*Agoro et al., 2018*). It has also been shown that erythropoietin-mediated erythropoietic drive could suppress the inductive effects of iron or inflammation on hepcidin expression by inhibiting SMAD4 and STAT3 signaling (*Huang et al., 2009*). Moreover, a previous study reported that the expression of hepcidin under thalassemic conditions is concurrently regulated by both systemic iron loading and ineffective erythropoiesis, which have opposing effects on hepcidin (*Gardenghi et al., 2007*). Therefore, different degrees of ineffective erythropoiesis and iron loading could lead to different hepcidin levels in these patients.

Despite several studies in the effects of inflammation on iron homeostasis, the responses of iron homeostasis to acute inflammation under thalassemic condition have not been elucidated. Moreover, it has not been addressed whether such responses would be altered

by the presence of systemic iron overload. The present study, therefore, aimed to explore the effects of LPS administration, a model of acute inflammation, on the expression of both hepcidin and iron transport molecules in thalassemic mouse model with and without parenteral iron loading. The information acquired from this study will provide better understanding of iron homeostasis in thalassemic patients with concurrent acute inflammation/infection.

## MATERIALS & METHODS

### Animal care and treatment

The present study utilized heterozygous β-knockout ($Hbb^{th3/+}$) mice that harbored heterozygous deletion of both murine adult β-globin genes ($\beta major$ and $\beta minor$). This mouse model demonstrates comparable features to thalassemia intermedia, including anemia, hepatosplenomegaly, ineffective erythropoiesis, and extramedullary hematopoiesis (*Yang et al., 1995*).

Male 8-to-12-week-old $Hbb^{th3/+}$ and wild type (WT) mice littermates under C57BL/6J background were obtained from the Thalassemia Research Center, Institute of Molecular Biosciences, Mahidol University. All animals were given routine feeding with standard rodent chow (C.P. mice feed 082G containing 180 mg/kg of iron; Perfect Companion Group, Thailand) and water ad libitum. The temperature and humidity were maintained at 25 ± 2 °C and 55 ± 10%, respectively, with a 12-h light/dark cycle and clean conventional housing system. The mice were subjected to parenteral iron loading and/or acute inflammatory induction (five mice per group).

For parenteral iron loading, WT and $Hbb^{th3/+}$ mice were intramuscularly injected with 5 mg of iron dextran (Sigma–Aldrich, St. Louis, MO, USA) once daily for two consecutive days (a total dose of 10 mg). After a 2-week equilibration, acute inflammation was induced by an intraperitoneal injection of a single dose of 1 μg/g body weight of lipopolysaccharide (LPS) (Sigma–Aldrich, St. Louis, MO, USA). Control groups for both iron loading and acute inflammation received equal volume(s) of saline solution. The mice were sacrificed under Pentobarbital-induced anesthesia by exsanguination at 6 h after LPS (or saline) injection. Blood samples were collected by cardiac puncture and tissue samples (liver, spleen and duodenum) were snap frozen and stored at −20 and −80 °C, respectively. All animal studies were approved by Institute of Molecular Biosciences Animal Care and Use Committee (IMB-ACUC) of Mahidol University, Thailand (COA. NO. MUMB-ACUC 2017/003). All experiments were performed at Mahidol University, Thailand in accordance with the approved protocol and local regulations.

### Determination of hematological and iron parameters

Hematological parameters were analyzed using an automated hematological analyzer (Mindray, Shenzhen, China). Serum iron concentration was determined using a QuantiChrom iron assay kit (BioAssay System, Hayward, CA, USA) according to the manufacturer's protocol.

Tissue non-heme iron contents in the liver and spleen were measured by a modification of the method of *Foy et al. (1967)* as described by *Simpson & Peters (1990)*.

**Table 1 Sequence of gene-specific primers.**

| Gene product | Forward primer | Reverse primer |
|---|---|---|
| *Actb* (β-actin) | 5′-CAGCCTTCCTTCTTGGGTA-3′ | 5′-TTTACGGATGTCAACGTCACAC-3′ |
| *Bmp6* (BMP6) | 5′-GCCAACTACTGTGATGGAGAGTGTT-3′ | 5′-CTCGGGATTCATAAGGTGGACCA-3′ |
| *Crp* (CRP) | 5′-AGCTTCTCTCGGACTTTTGGT-3′ | 5′-GGTGTTCAGTGGCTTCTTTGA-3′ |
| *Cybrd1* (DCYTB) | 5′-TTTGTCCTGAAACACCCCTC-3′ | 5′-AGAAGGCCCAGCGTATTTGT-3′ |
| *Fam132b* (ERFE) | 5′-TCCTCTATCTACAGGCAGGAC-3′ | 5′-ACTGCGTACCGTGAGGGA-3′ |
| *Hamp* (Hepcidin) | 5′-CAGGGCAGACATTGCGATAC-3′ | 5′-GTGGCTCTAGGCTATGTTTTGC |
| *Il6* (IL-6) | 5′-TCTAATTCATATCTTCAACCAAGAGG-3′ | 5′-TGGTCCTTAGCCACTCCTTC-3′ |
| *Slc11a2* (DMT1) (+IRE isoform) | 5′-TTCTACTTGGGTTGGCAGTGTT-3′ | 5′-CAGCAGGACTTTCGAGATGC-3′ |
| *Slc40a1* (FPN1) | 5′-ATCCCCATAGTCTCTGTCAGC-3′ | 5′-CAGCAACTGTGTCACCGTCA-3′ |
| *Tmprss6* (TMPRSS6) | 5′-ACTCTTGAAGATGCCGAGATG-3′ | 5′-GCAGCTTCCTCTCCATCACC-3′ |

## Quantitative RT-PCR

RNA was extracted from the liver, spleen, and duodenal samples using TRIzol reagent (Ambion, Austin, TX, USA). RNA purity was measured using a Nanophotometer (Implen GmbH, Munich, Germany), with an acceptable A260/280 ratio of 1.8 to 2.2, and an acceptable A260/230 ratio of >1.7. Complementary DNA (cDNA) was synthesized using a Tetro cDNA synthesis kit (Bioline, Taunton, MA, USA) according to the manufacturer's protocol and stored at −20 °C. Quantitative RT-PCR was performed using a CFX96 Thermal Cycler (Bio-Rad Laboratories, Irvine, CA, USA) using SYBR dye (Roche Diagnostics, Mannheim, Germany). PCR reactions consisted of an initial denaturation at 95 °C for 10 min, followed by 40 cycles of 95 °C (denaturation) for 10 s, 59 °C (annealing) for 45 s, and 72 °C (extension) for 30 s. Each PCR reaction was assayed in triplicate. Melting curve analysis was performed to confirm the specificity of the PCR reactions. Gene expression was normalized to β-actin (*Actb*) expression. Relative mRNA expression is presented as fold change compared to the expression in WT mice under basal condition (no iron dextran or LPS treatment) as obtained by the $2^{-\Delta\Delta CT}$ method (*Livak & Schmittgen, 2001*). The sequence of gene-specific primers is listed in Table 1.

## Statistical analysis

All data are expressed as mean ± standard error of the mean (SEM). Comparisons between different groups were performed using Kruskal–Wallis test with pairwise Mann–Whitney U test. The acquired *P* values were subsequently adjusted using the Bonferroni correction (*Lee & Lee, 2018*). All analyses were performed using SPSS software 16 (SPSS Inc., Chicago, IL, USA). An adjusted *P*-value less than 0.01 was considered statistically significant.

## RESULTS

### LPS exerted similar effects on most iron parameters in both *Hbb*[th3/+] and WT mice

*Hbb*[th3/+] mice displayed abnormal hematological parameters, including significantly reduced hemoglobin (Hb) and mean corpuscular hemoglobin (MCH), a marginally decreased hematocrit (Hct) along with a significantly increased red cell distribution width

**Table 2 Hematological parameters of wild type (WT) and thalassemic ($Hbb^{th3/+}$) mice treated with saline (Saline), lipopolysaccharide (LPS), iron dextran (Fe), or both iron dextran and LPS (Fe + LPS).**

| Hematological parameters | WT | | | | $Hbb^{th3/+}$ | | | |
|---|---|---|---|---|---|---|---|---|
| | Saline | LPS | Fe | Fe + LPS | Saline | LPS | Fe | Fe + LPS |
| RBC count ($10^6$/µL) | 4.82 ± 0.28 | 5.79 ± 0.67 | 6.33 ±0.37 | 7.86 ± 0.09[a,b] | 4.41 ± 0.43 | 3.80 ± 0.33 | 5.72 ± 0.22 | 5.88 ± 0.23 |
| Hemoglobin (g/dL) | 7.92 ± 0.45 | 9.66 ± 0.83 | 10.64 ± 0.59 | 13.20 ± 0.14[a,b] | 5.14 ± 0.45[a] | 5.48 ± 0.54 | 6.60 ± 0.25 | 6.80 ± 0.26 |
| Hematocrit (%) | 35.66 ± 1.92 | 39.82 ± 2.99 | 32.80 ± 1.76 | 39.36 ± 0.38[b] | 25.40 ± 2.13 | 26.38 ± 3.06 | 21.40 ± 0.77 | 21.40 ± 0.79 |
| MCV (fL) | 74.16 ± 0.92 | 70.12 ± 3.64 | 51.86 ± 0.51[a] | 50.08 ± 0.30[a] | 58.88 ± 4.79 | 68.60 ± 3.06 | 37.42 ± 0.41[c] | 36.50 ± 0.28[c] |
| MCH (pg) | 16.46 ± 0.16 | 16.96 ± 0.86 | 16.82 ± 0.10 | 16.84 ± 0.09 | 11.72 ± 0.19[a] | 14.62 ± 1.29 | 11.56 ± 0.02 | 11.54 ± 0.05 |
| MCHC (g/dL) | 22.20 ± 0.32 | 24.34 ± 1.29 | 32.48 ± 0.36[a] | 33.60 ± 0.27[a] | 20.38 ± 1.57 | 21.60 ± 2.47 | 30.94 ± 0.32[c] | 31.66 ± 0.20[c] |
| RDW (%) | 24.44 ± 0.83 | 24.34 ± 0.48 | 15.48 ± 1.13[a] | 13.42 ± 0.23[a] | 43.92 ± 1.44[a] | 35.18 ± 4.12 | 39.16 ± 0.85 | 36.10 ± 0.60[c] |
| Reticulocyte (%) | 1.04 ± 0.35 | 0.80 ± 0.38 | 0.06 ± 0.02[a] | 0.12 ± 0.07 | 0.78 ± 0.14 | 0.90 ± 0.18 | 0.18 ± 0.10 | 0.10 ± 0.04[c] |

Notes:
[a] Adjusted *P*-value < 0.01 compared with WT-Saline.
[b] Adjusted *P*-value < 0.01 compared with WT-Fe.
[c] Adjusted *P*-value < 0.01 compared with $Hbb^{th3/+}$-Saline.
RBC, red blood cell; MCV, mean corpuscular volume; MCH, mean corpuscular hemoglobin; MCHC, mean corpuscular hemoglobin concentration; RDW, red cell distribution width.
Data are expressed as mean ± SEM (*n* = 5/group). Statistical analysis was performed using Kruskal–Wallis test with pairwise Mann–Whitney U test. The acquired *P* values were subsequently adjusted using the Bonferroni correction.

**Table 3 Iron parameters of wild type (WT) and thalassemic ($Hbb^{th3/+}$) mice treated with saline (Saline), lipopolysaccharide (LPS), iron dextran (Fe), or both iron dextran and LPS (Fe + LPS).**

| Iron parameters | WT | | | | $Hbb^{th3/+}$ | | | |
|---|---|---|---|---|---|---|---|---|
| | Saline | LPS | Fe | Fe + LPS | Saline | LPS | Fe | Fe + LPS |
| Serum iron (µL) | 27.05 ± 1.65 | 16.58 ± 2.08 | 46.67 ± 4.66 | 29.76 ± 1.14[b] | 19.65 ± 2.44 | 9.77 ± 1.08[c] | 31.01 ± 2.90 | 17.23 ± 2.15[d] |
| Liver non-heme iron (nmole/mg wet weight) | 2.60 ± 0.26 | 2.31 ± 0.13 | 62.36 ± 11.18[a] | 65.56 ± 8.40[a] | 4.57 ± 0.53 | 4.43 ± 0.90 | 74.56 ± 4.85[c] | 63.89 ± 2.43[c] |
| Spleen non-heme iron (nmole/mg wet weight) | 8.32 ± 0.89 | 5.79 ± 0.48[a] | 43.27 ± 7.32[a] | 66.80 ± 8.47[a] | 29.41 ± 1.90[a] | 25.09 ± 0.96 | 46.95 ± 4.30 | 42.44 ± 5.56 |

Notes:
[a] Adjusted *P*-value < 0.01 compared with WT-Saline.
[b] Adjusted *P*-value < 0.01 compared with WT-Fe.
[c] Adjusted *P*-value < 0.01 compared with $Hbb^{th3/+}$-Saline.
[d] Adjusted *P*-value < 0.01 compared with $Hbb^{th3/+}$-Fe.
Data are expressed as mean ± SEM (*n* = 4–5/group). Statistical analysis was performed using Kruskal–Wallis test with pairwise Mann–Whitney U test. The acquired *P* values were subsequently adjusted using the Bonferroni correction.

(RDW), which corresponded with thalassemic phenotype (Table 2). Parenteral iron administration in both WT and $Hbb^{th3/+}$ mice led to an increase in mean corpuscular hemoglobin concentration (MCHC), while mean corpuscular volume (MCV), RDW, and reticulocyte count were reduced (Table 2). In WT mice with parenteral iron loading, LPS treatment was associated with significantly increased RBC count, hemoglobin and hematocrit (Table 2). Similar responses were also observed in LPS-treated WT mice under control condition (no iron dextran injection), but the differences did not reach a statistically significant level. In contrast, hematological parameters of $Hbb^{th3/+}$ mice under both control and iron loading conditions were mostly unaffected by LPS administration.

Under basal condition (no iron dextran or LPS administration), tissue iron overload was observed in $Hbb^{th3/+}$ mice as evidenced by increased liver and spleen non-heme iron levels compared to WT counterpart (Table 3). Iron dextran administration was associated

with increased serum iron levels, as well as increased liver and spleen non-heme iron content in both phenotypes (Table 3). Notably, LPS treatment was associated with a reduction in serum iron levels in WT and $Hbb^{th3/+}$ mice under both control and parenteral iron loading conditions (Table 3). A significant reduction in spleen non-heme iron content upon LPS injection was found only in WT mice under control condition (Table 3). Otherwise, tissue non-heme iron content in WT and $Hbb^{th3/+}$ mice under both conditions was generally unaffected by LPS.

## Induction of liver IL-6 and hepcidin by LPS was influenced by thalassemia or iron status

Liver interleukin 6 and C-reactive protein (CRP) mRNA expression were determined at 6 hours after LPS administration to confirm inflammatory induction by LPS treatment. We found significantly increased IL-6 mRNA levels in LPS-treated WT and $Hbb^{th3/+}$ mice compared to their control counterparts (Fig. 1A) suggesting that acute inflammation was successfully induced. Although iron dextran injection alone did not affect the mRNA expression of IL-6, the magnitude of IL-6 induction by LPS in both phenotypes was higher under parenteral iron loading condition (95 folds in WT and 194 folds in $Hbb^{th3/+}$) than control (no iron dextran treatment) condition (11 folds in WT and 17 folds in $Hbb^{th3/+}$). Furthermore, liver CRP mRNA expression in WT and $Hbb^{th3/+}$ mice under both conditions was marginally increased upon LPS administration (Fig. 1B).

Under basal condition, a trend toward decreased liver hepcidin mRNA expression was noted in $Hbb^{th3/+}$ mice compared to WT mice (Fig. 1C). The administration of either iron dextran or LPS was associated with increased hepcidin mRNA expression in both WT and $Hbb^{th3/+}$ mice. Interestingly, the results in WT mice revealed that the extent of hepcidin induction by LPS was more pronounced under parenteral iron condition (4.6-fold induction; WT—Fe+LPS vs WT—Fe ; Fig. 1C) than control condition (2.4-fold induction; WT—LPS vs WT—Saline; Fig. 1C). In iron dextran-treated $Hbb^{th3/+}$ mice, liver hepcidin mRNA expression was unaffected by LPS administration ($Hbb^{th3/+}$—Fe vs $Hbb^{th3/+}$—Fe+LPS; Fig. 1C).

## Parenteral iron loading altered the responses of splenic ERFE mRNA expression to LPS

Quantitative RT-PCR revealed a significant increase in splenic ERFE mRNA expression and a trend toward increased liver BMP6 mRNA expression in $Hbb^{th3/+}$ mice compared to WT mice under basal condition (Figs. 2A and 2B). Iron dextran injection significantly induced liver BMP6 mRNA expression in both WT and $Hbb^{th3/+}$ mice (Fig. 2B), whereas the mRNA expression of splenic ERFE and liver TMPRSS6 was not affected by parenteral iron loading (Figs. 2A and 2C).

LPS administration significantly suppressed liver BMP6 mRNA expression in WT and $Hbb^{th3/+}$ mice under both conditions (Figs. 2B). Upon LPS treatment, liver TMPRSS6 mRNA levels of both WT and $Hbb^{th3/+}$ mice under control condition was also significantly decreased, however, such responses were lessened under parenteral iron loading particularly in $Hbb^{th3/+}$ mice (Figs. 2C). On the contrary, splenic ERFE mRNA expression

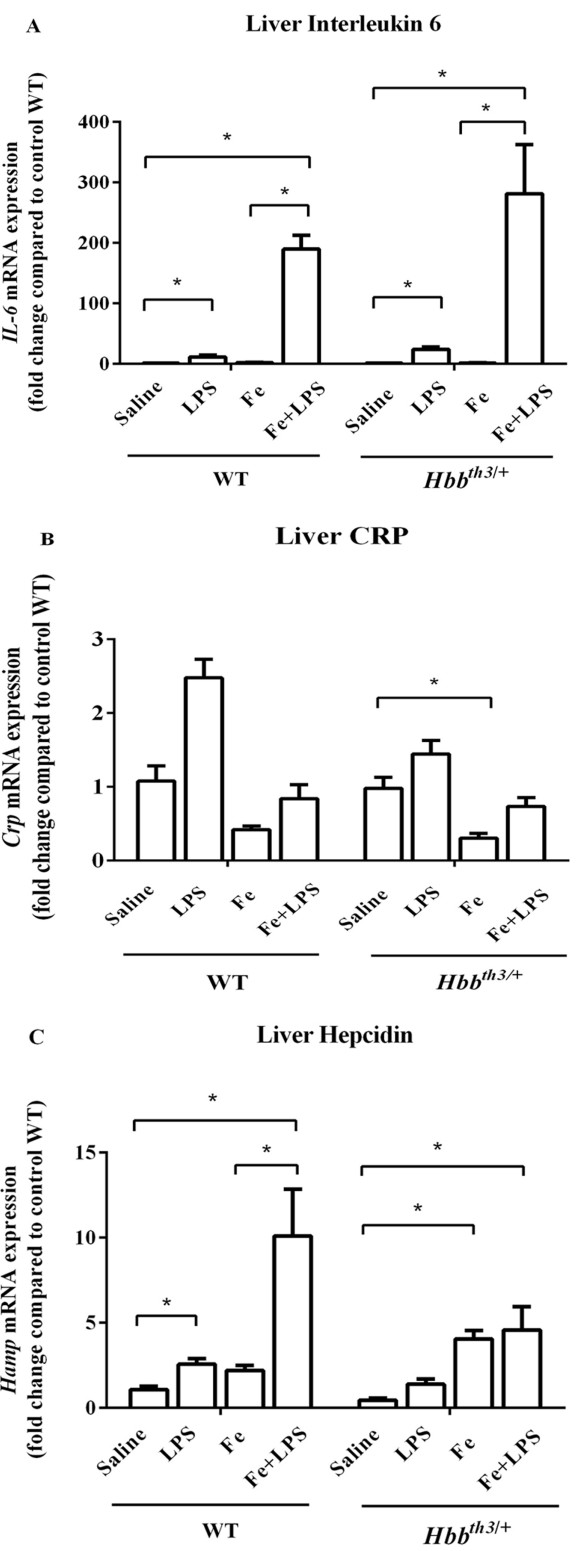

**Figure 1 Effects of LPS on the mRNA expression of interleukin 6, C-reactive protein and hepcidin in the liver of wild type and thalassemic mice with/without parenteral iron loading.** The mRNA expression of (A) interleukin 6 (IL-6), (B) C-reactive protein (CRP) and (C) hepcidin in the liver of wild type (WT) and thalassemic ($Hbb^{th3/+}$) mice treated with iron dextran/saline followed by LPS/saline

**Figure 1** (continued)
administration. Tissue samples were collected at 6 hours after LPS/saline injection. Gene expression was normalized to β-actin (*Actb*) expression. Data are presented as mean and SEM of the fold change compared to saline-treated WT mice (WT-Saline) (*n* = 5 per group). Statistical analysis was performed using Kruskal–Wallis test with pairwise Mann–Whitney U test. The acquired *P* values were subsequently adjusted using the Bonferroni correction (*adjusted *P*-value < 0.01).

was downregulated by LPS injection only under parenteral iron loading condition particularly in $Hbb^{th3/+}$ mice (Fig. 2A).

## The effects of LPS on the mRNA expression of major iron transporters were present in thalassemia mice even under parenteral iron loading condition

The mRNA expression of major iron transport molecules, namely DCYTB, DMT1 and FPN1, in the liver, spleen, and duodenum was determined by real-time quantitative RT-PCR. Under basal condition, the expression of DMT1 in the liver did not differ between WT and $Hbb^{th3/+}$ mice (Fig. 3A) while liver FPN1 mRNA expression was marginally higher in $Hbb^{th3/+}$ mice (Fig. 3B). Moreover, $Hbb^{th3/+}$ mice demonstrated significant induction of DMT1 and FPN1 mRNA expression in the spleen compared to WT mice (Figs. 3C and 3D). Upon iron dextran injection, FPN1 mRNA expression was significantly induced in the liver of both WT and $Hbb^{th3/+}$ mice (Fig. 3B). In contrast, DMT1 mRNA levels in the liver and spleen as well as FPN1 mRNA levels in the spleen of both phenotypes were not affected by parenteral iron loading (Figs. 3A, 3C and 3D).

The administration of LPS significantly suppressed FPN1 mRNA expression in the liver and spleen of WT and $Hbb^{th3/+}$ mice under both control condition and parenteral iron loading condition (Figs. 3B and 3D). As for DMT1, LPS treatment was associated with increased liver DMT1 mRNA levels and decreased splenic DMT1 mRNA levels, however, the changes were statistically significant only in $Hbb^{th3/+}$ mice with parenteral iron loading (Figs. 3A and 3C).

With regards to the duodenum, increased mRNA levels of DCYTB, DMT1 and FPN1 in $Hbb^{th3/+}$ mice compared to WT mice were observed under basal condition (Figs. 4A–4C). The mRNA expression of these iron transport molecules was not affected by iron dextran treatment apart from a trend toward DCYTB suppression in WT mice. Upon LPS administration under both control and parenteral iron loading conditions, the mRNA expression of DCYTB, DMT1 and FPN1 was generally downregulated in both WT and $Hbb^{th3/+}$ mice; however, the responses were more pronounced in $Hbb^{th3/+}$ mice.

## DISCUSSION

The current study was conducted to determine iron homeostatic responses of thalassemic mouse model to acute inflammation in the presence or absence of parenteral iron loading. Thalassemic phenotype of the mouse model was evidenced by the presence of hypochromic microcytic anemia along with parenchymal iron loading. In this study, parenteral iron loading was induced by intramuscular administration of iron dextran

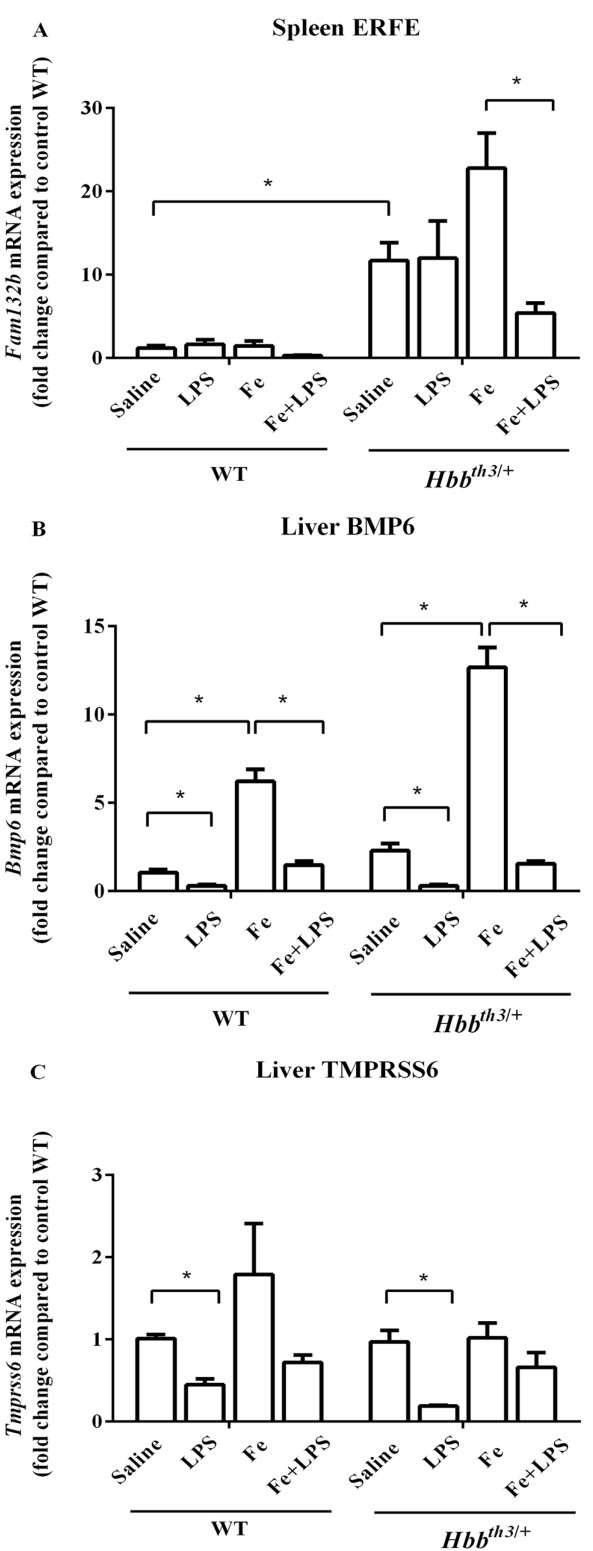

**Figure 2 Effects of LPS on the mRNA expression of upstream regulators of hepcidin in wild type and thalassemic mice with/without parenteral iron loading.** The mRNA expression of (A) spleen ERFE, (B) liver BMP6 and (C) liver TMPRSS6 in wild type (WT) and thalassemic ($Hbb^{th3/+}$) mice treated with iron dextran/saline followed by LPS/saline administration. Tissue samples were collected at 6 hours after

**Figure 2** (continued)

LPS/saline injection. Gene expression was normalized to β-actin (*Actb*) expression. Data are presented as mean and SEM of the fold change compared to saline-treated WT mice (WT-Saline) (*n* = 5 per group). Statistical analysis was performed using Kruskal–Wallis test with pairwise Mann–Whitney U test. The acquired *P* values were subsequently adjusted using the Bonferroni correction (*adjusted *P*-value < 0.01). 

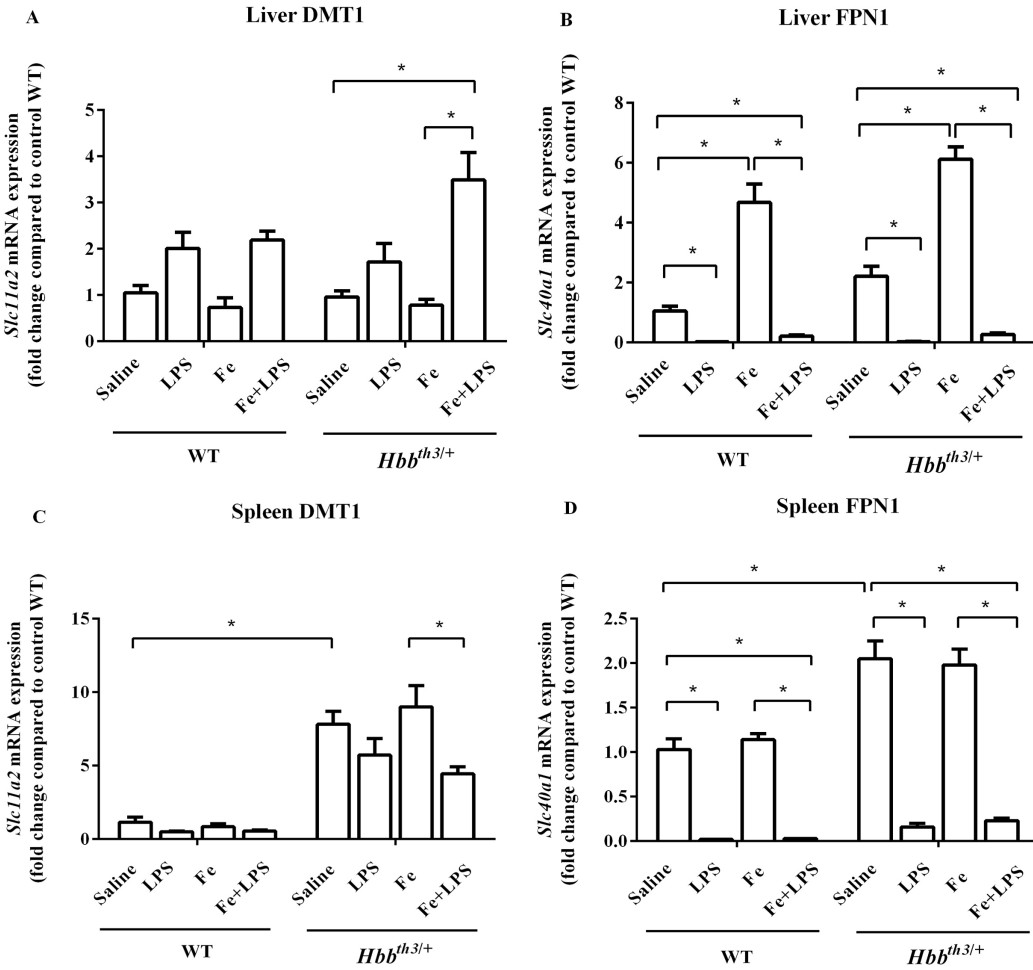

**Figure 3 Effects of LPS on the mRNA expression of DMT1 and FPN1 in the liver and spleen of wild type and thalassemic mice with/without parenteral iron loading.** The mRNA expression of (A) liver DMT1, (B) liver ferroportin (FPN1), (C) spleen DMT1 and (D) spleen ferroportin (FPN1) in wild type (WT) and thalassemic (*Hbb*[th3/+]) mice treated with iron dextran/saline followed by LPS/saline administration. Tissue samples were collected at 6 hours after LPS/saline injection. Gene expression was normalized to β-actin (*Actb*) expression. Data are presented as mean and SEM of the fold change compared to saline-treated WT mice (WT-Saline) (*n* = 4–5 per group). Statistical analysis was performed using Kruskal–Wallis test with pairwise Mann–Whitney U test. The acquired *P* values were subsequently adjusted using the Bonferroni correction (*adjusted *P*-value < 0.01).

which has also been utilized in previous reports (*Atanasova et al., 2004*; *Montosi et al., 2005*; *Laftah et al., 2005*; *Dong et al., 2020*). Systemic iron overload was demonstrated by the increased iron levels in serum, liver, and spleen. Acute inflammation was induced by

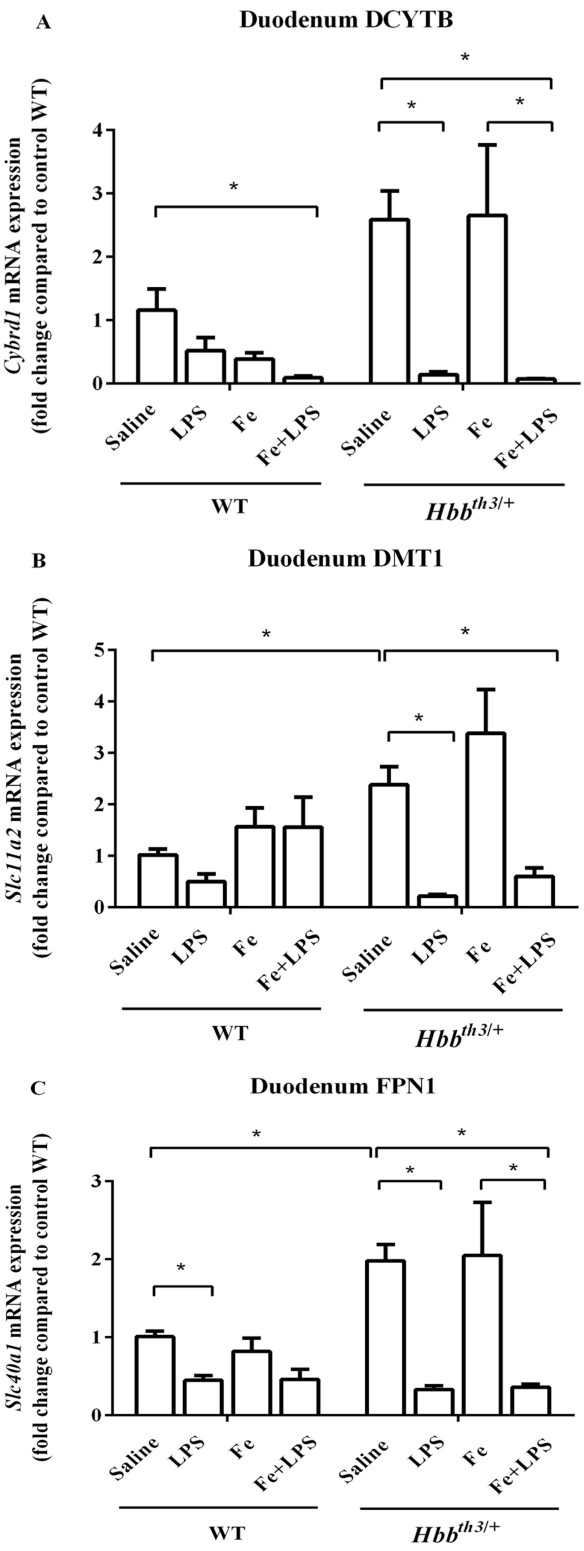

**Figure 4 Effects of LPS on the mRNA expression of iron transport molecules in the duodenum of wild type and thalassemic mice with/without parenteral iron loading.** The mRNA expression of (A) DCYTB, (B) DMT1 and (C) ferroportin (FPN1) in the duodenum of wild type (WT) and thalassemic ($Hbb^{th3/+}$) mice treated with iron dextran/saline followed by LPS/saline administration. Tissue samples

**Figure 4** (continued)
were collected at 6 hours after LPS/saline injection. Gene expression was normalized to β-actin (*Actb*) expression. Data are presented as mean and SEM of the fold change compared to saline-treated WT mice (WT-Saline) (*n* = 5 per group). Statistical analysis was performed using Kruskal–Wallis test with pairwise Mann–Whitney U test. The acquired *P* values were subsequently adjusted using the Bonferroni correction (*adjusted *P*-value < 0.01).              

intraperitoneal injection of 1 μg/g body weight of lipopolysaccharide (LPS). This sublethal dose of LPS has widely been utilized in previous studies (*Fillebeen et al., 2018*; *Huang et al., 2009*; *De Domenico et al., 2010*; *Krijt et al., 2006*; *Meynard et al., 2009*; *Latour et al., 2017*). Acute inflammatory induction was achieved as suggested by the upregulation of liver IL-6 and CRP mRNA expression at 6 hours after LPS injection. Notably, the magnitude of LPS-mediated IL-6 induction was more pronounced in the presence of iron dextran treatment, while such response was not affected by thalassemia. In agreement with our results, previous studies in mice and mouse macrophage cell line reported that the effects of iron status on LPS-induced IL-6 expression were enhanced by pretreatment with iron (*Layoun & Santos, 2012*; *Hoeft et al., 2017*). Interestingly, it has been reported that increased levels of intracellular labile iron in inflammatory cells could result in the alteration of mitochondrial homeostasis leading to increased cytokine response to LPS challenge (*Hoeft et al., 2017*). On the contrary, another study revealed that LPS-induced pro-inflammatory responses of bone marrow-derived macrophages were impaired by ferric ammonium citrate treatment through the reduction of NF-kappa B p65 nuclear translocation (*Agoro et al., 2018*). Such discrepant findings could be due to the differences in cell types, study models, method of iron treatment, and dosages of iron and LPS.

The expression of hepcidin is regulated by several factors including IL-6, iron levels and ineffective erythropoiesis. Notably, the crosstalk between the BMP-SMAD and JAK-STAT pathways relative to the regulation of hepcidin expression has previously been reported (*Besson-Fournier et al., 2012*; *Canali et al., 2016*; *Fillebeen et al., 2018*; *Gallitz et al., 2018*; *Verga Falzacappa et al., 2008*; *Yu et al., 2008*; *Steinbicker et al., 2011*; *De Domenico et al., 2010*). Therefore, we further explored the responses of hepcidin to iron dextran and/or LPS challenge in WT and thalassemic mice. As expected, a suppressive effect of thalassemia as well as inductive effects of iron loading and LPS on hepcidin expression were found. In agreement with a previous study (*Hoeft et al., 2017*), our study demonstrated that iron dextran injection and LPS challenge synergistically induced hepcidin induction in the liver of WT mice. These findings suggest that hepcidin could concurrently and synergistically be induced by iron and inflammation. In contrast to WT mice, we noted that LPS injection failed to increase liver hepcidin mRNA levels in $Hbb^{th3/+}$ mice pretreated with iron dextran. We speculated that this blunted effect of LPS in iron dextran-treated $Hbb^{th3/+}$ mice might be caused by the altered expression of upstream hepcidin modulator(s) in thalassemic mice.

It has been proposed that the levels of hepcidin expression are determined by the relative strength and the duration of each individual hepcidin regulatory signal

(*Stoffel et al., 2019*; *Huang et al., 2009*). We, therefore, determined the expression of major molecules involved in hepcidin regulation in response to iron status and ineffective erythropoiesis. We observed suppressive effects of LPS on the expression of BMP6 and TMPRSS6 in both WT and $Hbb^{th3/+}$ mice. Similar results have previously been reported not only in WT mice, but also in hepcidin knockout mice that also exhibit systemic iron overload (*Deschemin & Vaulont, 2013*). In our study, it is noteworthy that the extent of TMPRSS6 downregulation by inflammation in both WT and $Hbb^{th3/+}$ mice was lessened in the presence of parenteral iron loading, which coincided with a remarkable induction of IL-6. As HJV was required for LPS-mediated hepcidin induction (*Fillebeen et al., 2018*), it is possible that the suppression of TMPRSS6 by inflammation would serve to adjust a proper amount of membrane-bound HJV in order to facilitate the appropriate level of hepcidin induction in response to inflammation. We also speculate that the attenuation of TMPRSS6 mRNA suppression upon LPS injection under parenteral iron loading condition might contribute to the blunted hepcidin response to LPS in iron dextran-treated $Hbb^{th3/+}$ mice. Further studies should be performed to determine the activities of the BMP-SMAD and JAK-STAT pathways at the protein level in order to confirm this speculation.

Regarding erythroid regulators, splenic expression of ERFE was suppressed at 6 hours after LPS administration only under parenteral iron loading condition particularly in $Hbb^{th3/+}$ mice. Interestingly, a study in critically ill patients found that serum ERFE levels were decreased over time in patients with sepsis, and in patients developing anemia of inflammation (*Boshuizen et al., 2018*). However, the information regarding the effects of acute inflammation on ERFE expression especially under systemic iron loading condition is quite limited. Further studies are required to confirm whether iron status affects the response of ERFE to acute inflammation.

Next, we examined the expression of DCYTB, DMT1 and FPN1 in the duodenum, liver, and spleen. In general, WT and $Hbb^{th3/+}$ mice exhibited similar pattern of responses to LPS challenge but the magnitudes of some changes slightly differed according to the presence of thalassemia or parenteral iron loading. We observed that the altered mRNA expression of DMT1 in the liver and spleen upon LPS treatment was more remarkable in iron dextran-treated $Hbb^{th3/+}$ mice. With regards to FPN1, our study showed that LPS injection was associated with decreased FPN1 mRNA levels in the duodenum, liver, and spleen of WT and $Hbb^{th3/+}$ mice under both control and parenteral iron loading conditions. Moreover, we observed that LPS could override the effect of iron loading on liver FPN1 expression in both phenotypes. It is also noteworthy that the downregulation of duodenal iron transport molecules by LPS was more pronounced in $Hbb^{th3/+}$ mice and such responses in $Hbb^{th3/+}$ mice were not affected by parenteral iron loading.

The downregulation of FPN1 in the duodenum, liver, and spleen would reduce the entry of iron into the circulation. Correspondingly, serum iron levels were decreased upon LPS challenge in both WT and $Hbb^{th3/+}$ mice—even in the presence of parenteral iron loading. Notably, LPS injection in iron dextran-treated $Hbb^{th3/+}$ mice resulted in transcriptional alteration of iron transport molecules and reduced serum iron levels

Table 4 Summary of the results regarding the responses of iron homeostasis to LPS administration in wild type (WT) and thalassemic ($Hbb^{th3/+}$) mice under control and parenteral iron loading conditions.

| | Control condition | | Parenteral iron loading condition | |
|---|---|---|---|---|
| | WT | $Hbb^{th3/+}$ | WT | $Hbb^{th3/+}$ |
| **Iron parameters** | | | | |
| Serum iron | (↓) | ↓ | ↓ | ↓ |
| Liver non-heme iron | – | – | – | – |
| Spleen non-heme iron | ↓ | – | – | – |
| **Inflammatory markers** | | | | |
| IL-6 | ↑ | ↑ | ↑ | ↑ |
| CRP | (↑) | (↑) | (↑) | (↑) |
| **Hepcidin and its upstream regulators** | | | | |
| ERFE | – | – | (↓) | ↓ |
| BMP6 | ↓ | ↓ | ↓ | ↓ |
| TMPRSS6 | ↓ | ↓ | (↓) | (↓) |
| Hepcidin | ↑ | (↑) | ↑ | – |
| **Iron transport molecules** | | | | |
| Liver DMT1 | (↑) | (↑) | (↑) | ↑ |
| Liver FPN1 | ↓ | ↓ | ↓ | ↓ |
| Spleen DMT1 | (↓) | (↓) | (↓) | ↓ |
| Spleen FPN1 | ↓ | ↓ | ↓ | ↓ |
| Duodenum DCYTB | (↓) | ↓ | (↓) | ↓ |
| Duodenum DMT1 | (↓) | ↓ | – | (↓) |
| Duodenum FPN1 | ↓ | ↓ | (↓) | ↓ |

**Notes:**
↑ A significant increase.
↓ A significant decrease.
(↑) A marginal increase or a trend toward an increase.
(↓) A marginal decrease or a trend toward a decrease.
– No effect.
IL-6, interleukin-6; CRP, C-reactive protein; ERFE, erythroferrone; BMP6, bone morphogenetic protein 6; TMPRSS6, matriptase-2; DMT1, divalent metal transporter 1; FPN1, ferroportin; DCYTB, duodenal cytochrome b.

despite the unaltered hepcidin expression. Therefore, such responses should be, at least partly, hepcidin-independent. In agreement, a previous study demonstrated that suppression of DCYTB and DMT1 in the duodenum, as well as hypoferremia, could be induced by LPS in hepcidin knockout mice (*Deschemin & Vaulont, 2013*). In addition, inflammatory induction via stimulation of Toll-like receptor 2 has been shown to induce transcriptional suppression of FPN1 and subsequent hypoferremia independent of hepcidin (*Guida et al., 2015*).

According to the results of the present study summarized in Table 4, our study demonstrated that the hypoferremic response to LPS is maintained in $Hbb^{th3/+}$ mice under both control and parenteral iron loading conditions possibly in a hepcidin-independent manner through the transcriptional suppression of FPN1 and duodenal iron transport molecules.

## CONCLUSIONS

In summary, the present study demonstrated that inflammation could alter the expression of hepcidin and iron transport molecules, as well as lower serum iron levels in both WT and thalassemic mice—even under parenteral iron loading, at least partly, in a hepcidin-independent manner. Our study demonstrated that the hypoferremic response to acute inflammation is maintained in iron-loaded thalassemic mice. A similar response might be expected in thalassemic patients in response to inflammation or infection. As such, inflammatory status should be taken into account in the assessment of iron parameters in these patients. The limitations of the present study include the limited number of mice ($n = 5$) in each group. Additionally, the effects of acute inflammation on protein levels were not determined since this study focused mainly on responses at steady state mRNA levels. Therefore, further and broader studies with a larger sample size should be conducted to explore the expression of key molecules (e.g. IL-6, hepcidin, ERFE and FPN1) at the protein level. Moreover, the impact of chronic inflammation/infection on iron homeostasis and hematological parameters under iron-loaded thalassemic condition should be further examined.

## ACKNOWLEDGEMENTS

We would like to thank the Thalassemia Research Center, Institute of Molecular Biosciences, Mahidol University for supplying thalassemic mice, and the Department of Biochemistry, Faculty of Medicine Siriraj Hospital, Mahidol University for supplying research instruments.

### Funding

This work is supported by the Siriraj Research Fund, Faculty of Medicine Siriraj Hospital, Mahidol University, and a National Research University (NRU) scholarship, Thailand. Chanita Sanyear and Professor Dr. Punnee Butthep were supported by a Royal Golden Jubilee (RGJ) Ph. D. Programme Scholarship from the Thailand Research Fund (PHD/0052/2556). The funders had no role in study design, data collection and analysis, decision to publish, or preparation of the manuscript.

### Grant Disclosures

The following grant information was disclosed by the authors:
Faculty of Medicine Siriraj Hospital, Mahidol University, and a National Research University (NRU), Thailand.
Thailand Research: PHD/0052/2556.

### Competing Interests

The authors declare that they have no competing interests.

## Author Contributions

- Chanita Sanyear conceived and designed the experiments, performed the experiments, analyzed the data, prepared figures and/or tables, authored or reviewed drafts of the paper, and approved the final draft.
- Buraporn Chiawtada performed the experiments, prepared figures and/or tables, and approved the final draft.
- Punnee Butthep conceived and designed the experiments, prepared figures and/or tables, and approved the final draft.
- Saovaros Svasti conceived and designed the experiments, prepared figures and/or tables, and approved the final draft.
- Suthat Fucharoen conceived and designed the experiments, prepared figures and/or tables, and approved the final draft.
- Patarabutr Masaratana conceived and designed the experiments, performed the experiments, analyzed the data, prepared figures and/or tables, authored or reviewed drafts of the paper, and approved the final draft.

## Animal Ethics

The following information was supplied relating to ethical approvals (i.e., approving body and any reference numbers):

Institute of Molecular Biosciences Animal Care and Use Committee (IMB-ACUC) of Mahidol University, Thailand approved this research (COA. NO. MUMB-ACUC 2017/003).

## Data Availability

The raw data are available in the Supplemental File.

## Supplemental Information

Supplemental information for this article can be found online at http://dx.doi.org/10.7717/peerj.11367#supplemental-information.

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
