# Peer review of "The hypoferremic response to acute inflammation is maintained in thalassemia mice even under parenteral iron loading"

_PeerJ, doi:10.7717/peerj.11367_

## Round 0.1 · original submission · Major Revisions

Please address the thoughtful comments from our reviewers, as well as the following points:

1) a large number of comparisons was performed. A multiple-comparison adjustment for all p-values should therefore be performed (in all figures, as well as Tables 2 and 3.

2) Your discussion is somewhat hard to follow due to the large number of comparisons with previous work. Do you think those comparisons may also be depicted in a table where a summary of your results is compared with the previous data? If that is possible, I think the readability of your discussion would increase significantly, and that would also help to emphasize the importance of your results.

Reviewer 1 ·

Basic reporting

The work aims at verifying the effect of LPS and iron treatments, separately and together, on various indices of iron status in a mouse model of b-thalassemia and wild type controls. The results show that many indices respond similarly in the two types of mice, with the exception of Hepcidin, ERFE and DMT1 mRNAs. The work seems rather unfocused and the significance of the findings are unclear. This is evident even in the title with the obvious statement that "LPS alters iron homeostasis in mice....".

Experimental design

It is unclear why they they induced iron supplementation via intramuscular injection, that is an unusual way, since most studies use intravenous injections. Blood transfusions would have been a better model. The authors should indicate a reference for the treatment.

A sensitive index of inflammation other than IL6 is important to verify the success and extent of LPS treatment. And also to verify whether the intramuscular iron treatment induces inflammatory response. The liver CRP mRNA is one of the possible indices.

Validity of the findings

no comment

Additional comments

The discussion is far too long and unfocused. Rather than a summary of all previous studies on effect of iron and LPS in cells and mice, the discussion should deal with the comparison between WT and BKO mice and possibly provide an interpretation on the novel findings, such as the blunted hepcidin response and the higher levels of Fam132b and of Slc11A2 in BKO mice.

Reviewer 2 ·

Basic reporting

No comment.

Experimental design

No comment.

Validity of the findings

No comment.

Additional comments

In the manuscript the authors report the effect of iron loading and acute inflammation on iron homeostasis in wild-type and Hbbth3/+ (thalassemic) mice. They show that inflammation, independently of parenteral iron loading, modulated the expression of hepcidin and iron transport molecules in both WT and thalassemic mice. The hypoferremic response to acute inflammation was maintained in iron-loaded Hbbth3/+ mice and was, at least partially, independent on hepcidin regulation.

Minor comments:

1. Methods part in the abstract should be better described, while respecting the requirements of the journal – more information about treatments (iron loading, acute inflammation) instead of statistics.

2. Some abbreviations, while explained in the text, should also be defined in the abstract (e.g. FPN1, ERFE).

3. The authors are advised to use the designation Hbbth3/+ instead of BKO, both in the text, tables figures and legends.

---

## Round 0.2 · accepted · Accept

I am glad to accept your paper. I will ask the technical staff to change the title to your suggested "The hypoferremic response to acute inflammation is maintained in thalassemia mice even under parenteral iron loading".

Reviewer 1 ·

Basic reporting

no comment

Experimental design

no comment

Validity of the findings

no comment

Additional comments

The answers to the points I raised are satisfactory and the manuscript improved.
As a title, I think that the suggested one below gives a better description of the work than the present one.
“The hypoferremic response to acute inflammation is maintained in thalassemia mice even under parenteral iron loading”,